# Energy, Sodium, Sugar and Saturated Fat Content of New Zealand Fast-Food Products and Meal Combos in 2020

**DOI:** 10.3390/nu13114010

**Published:** 2021-11-10

**Authors:** Sally Mackay, Teresa Gontijo de Castro, Leanne Young, Grace Shaw, Cliona Ni Mhurchu, Helen Eyles

**Affiliations:** 1Department of Epidemiology and Biostatistics, The University of Auckland, Auckland 1010, New Zealand; t.castro@auckland.ac.nz (T.G.d.C.); grace.shaw@auckland.ac.nz (G.S.); h.eyles@auckland.ac.nz (H.E.); 2Nutrition Section, The University of Auckland, Auckland 1010, New Zealand; 3National Institute for Health Innovation, The University of Auckland, Auckland 1010, New Zealand; leanne.young@auckland.ac.nz (L.Y.); c.nimhurchu@auckland.ac.nz (C.N.M.); 4The George Institute for Global Health, Newtown, NSW 2042, Australia

**Keywords:** fast food, sodium, total sugars, population health, food environments, meal combos

## Abstract

This study aimed to benchmark the healthiness of the New Zealand (NZ) fast-food supply in 2020. There are currently no actions or policies in NZ regarding the composition, serving size and labeling of fast food. Data on serving size and nutrient content of products was collected from company websites and in-store visits to 27 fast-food chains. For each fast-food category and type of combo meal, medians and interquartile ranges were calculated for serving size and energy, sodium, total sugar, and saturated fat per serving. Nutrient contents/serving were benchmarked against the United Kingdom (UK) soft drinks levy sugar thresholds and targets for salt for away from home foods, the NZ daily intake guidelines for energy, sodium, and saturated fat, and the World Health Organization (WHO) recommendation for free sugars. Analyses were conducted for the 30.3% (n = 1772) of products with available nutrition information and for 176 meal combos. Most (n = 67; 91.8%) sugar-sweetened drinks would qualify for a UK soft drink industry levy and 47% (n = 1072) of products exceeded the relevant UK sodium target. Half of the meal combos provided at least 50.3% of the daily energy requirements and at least 88.6% of the maximum recommended intake of sodium. Fast-food products and combo meals in NZ contribute far more energy and negative nutrients to recommended daily intake targets than is optimal for good health. The NZ Government should set reformulation targets and serving size guidance to reduce the potential impact of fast- food consumption on the health of New Zealanders.

## 1. Introduction

New Zealand (NZ) has high rates of non-communicable disease with two of three adults and one in four children overweight or obese [1], and in 2017 unhealthy diets accounted for nearly 20% of illness and early death [2]. The average NZ diet is low in fruit, vegetables, wholegrains, legumes, nuts and seeds and contains an excess of foods high in sugar and sodium [1,2].

Fast-food is heavily marketed, cheap in comparison to other restaurant foods, convenient, accessible and palatable [3,4]. Fast-food has been independently associated with increased energy intakes and accelerating rates of weight gain and obesity [5,6]. In the United States, fast-food consumption has been associated with an additional 814 kJ of dietary energy per day and higher intakes of saturated fat and sodium [7]. Fast-food meals are generally characterized by large portion sizes, low levels of health-promoting nutrients such as fibre, and high levels of energy and adverse nutrients including saturated fat, added sugar and sodium [8]. Many chains offer combination meals (meal combos) in addition to individual items. These bundle unhealthy food options for a cheaper price and are a common tool used by the fast food industry to increase consumption [9].

In NZ, the percentage of the household food budget spent on restaurants and takeaways increased from 22% in 2000 to 27% in 2020 [10], and in 2019, 53% of adults bought a takeout meal from a fast-food or takeaway shop at least once a week [11]. Euromonitor trends for NZ from 2015 to 2020 show a 3.5% increase in foodservice value growth for limited-service restaurants with sales, and the number of fast-food outlets is forecast to continue to grow [12]. A previous assessment of the energy, serving size and sodium content of the NZ fast-food supply [13] reported an increase in the serving size of fast-food items from 2012 to 2016 and an increase in sodium and energy per serving, although there was no increase in product sodium or energy density.

Internationally there is a lack of agreed guidelines for portion or serving sizes or nutrients for fast-foods, and there is wide variation in serving sizes and nutrient contents of fast foods within and across countries [14]. However, the United Kingdom (UK) has government-led programs to reduce the energy, salt and sugar content in the “out-of-home” food sector [15]. The voluntary targets aim to reduce the levels of salt and sugar in the foods that contribute most to dietary intakes for UK adults. The mandatory UK “Soft Drinks Industry Levy” was introduced in 2018 to incentivize the industry to reduce the sugar content of soft drinks or pay a variable levy depending on the sugar content of the drink [16].

There are no government regulations in NZ related to fast-food composition targets, menu labeling of energy or nutrients [17], or to limit density of outlets. There is little nutrition information provided at the point of purchase to assist customers to purchase healthier options [18]. The NZ Government has not set food composition targets for any foods. The Heart Foundation has a voluntary HeartSAFE program focused on reducing sodium and sugar in low-cost, high-volume processed foods [19] but only four of 38 food categories have targets for foods that are consumed away from the home. In 2018, the NZ Ministers of Health and Primary Industries requested that the food industry convene a Food Industry Taskforce to show how they could contribute to obesity reduction. The resulting voluntary recommendations of the Taskforce related to fast food were to develop serving size ranges, best-practice portion guidance, education, providing nutrition information and to consider voluntary menu labeling [18] but there has been no indication of implementation. Annual cross-sectional surveys of all food and beverage products available for sale at fast-food chains in NZ are undertaken as part of data collection for the Nutritrack database [20]. Fast-food chains were defined according to Fleischhacker et al. [21] as restaurants providing food which is generally cheap, requires minimal preparation, and where no table service is provided.

This paper aims to benchmark the healthiness of products and combo meals available in the NZ fast-food supply in 2020 to provide recent evidence to inform effective policies and actions regarding reformulation and consumer information. To achieve this we: (i) assessed the sugar, salt, saturated fat and energy content of key fast-food product categories and meal combos and compared their nutrient content to national and international daily recommended intakes and; (ii) benchmarked selected fast-food product categories against accepted sodium and sugar targets. This is also the first study in NZ to assess the healthiness of meal combos provided by fast-food chains.

## 2. Materials and Methods

### 2.1. Data Sources

#### 2.1.1. Fast-Food Categories and Products

We conducted a cross-sectional survey of all food and beverage products available for sale at fast-food chains in NZ in 2020 [20]. All chains with ≥20 stores nationwide were selected (n = 27). Of these, twenty-two chains provided nutrition information for some or all products. Eleven chains were international chains and eleven were national chains. The following data were collected by trained fieldworkers between February and March in 2020, the year used for the current analysis: product name, serving and/or pack size, and nutrient information. Data were recorded directly from company websites. Visits to one large store representing each fast-food chain were also completed to capture any additional information not available on-line e.g., that on menu boards. Stores selected for visits were in Auckland, New Zealand’s largest city, and chosen based on size and location to reflect the largest product range possible. It is not mandatory for nutrition information to be available or displayed under the Australian and NZ Food Standards Code [17], and thus serving size and nutrient data were missing for some products.

#### 2.1.2. Fast-Food Combo Meals

We also created a database containing the nutrient composition of meal combos. This database used the data collected for the fast-food products (Section 2.2.1). We calculated the nutritional composition of meal combos by summing nutritional contents of individual products from this database. A combination of products was considered a meal combo if: (i) it offered two or more products, one of which may be a beverage; (ii) the meals were considered a main meal (e.g., burger, fries, soft drink) rather than a snack (e.g., muffin and coffee) and; (iii) the combo was promoted as a meal to be shared between a specific number of people (e.g., contained four burgers and four beverages). Some ‘party packs’ and pizza deals were not considered meals as it was not obvious how many people they would feed (e.g., three large pizzas, two large fries and 1.5 L soft drink).

### 2.2. Data Preparation

#### 2.2.1. Fast-Food Categories and Products

Of the 5840 products in the 2020 database, 30.4% were included in the analysis (n = 1772) and 59.8% (n = 3492) products were removed from the dataset as no information was available on serving size, package size and nutrients, with an additional 9.9% removed due to missing key nutrient data, implausible nutrient values, and/or no information on serving or package size. Food categories with less than 30 products were excluded except for ‘Fries’ (n = 25 products) because this is one of the most frequently consumed fast-food items [22], and there is little variation of products within the category. Those ‘Sides’ not available for sale separately were excluded (n = 118, 2.0%) from the analysis (Figure 1). Identical products with different serving sizes were retained as some nutrient targets [23] are set according to serving size. For products where the nutritional information was provided only per 100 g and the serving size was also available, the products` nutrition composition per serving was calculated.

Where applicable, we assessed the proportion of food items within the fast-food categories that exceeded the UK targets for sodium [23] and sugar [16]. This required further categorisation of some of the specific food items as the target to apply depended on certain ingredients (such as containing processed meat) or product size (fries < 8 mm or ≥8 mm in width).

#### 2.2.2. Fast-Food Combo Meals

The nutrient composition of meal combos in the database was calculated by combining per product nutrient values for existing individual products. A combination of products was considered a meal combo if: (i) it offered two or more products, one of which may be a beverage; (ii) the meals were considered a main meal (e.g., burger, fries, soft drink) rather than a snack (e.g., muffin and coffee) and; (iii) the combo was promoted as a meal to be shared between a specific number of people (e.g., contained four burgers and four beverages). Some ‘party packs’ and pizza deals were not considered meals as it was not obvious how many people they would feed (e.g., three large pizzas, two large fries and 1.5 L soft drink).

Meal combos were then categorized by type based on the presence of key common products e.g., burger, chicken, pie, sandwich, etc. Many of the combos offered the consumer a choice for one or more of the products within. When this occurred, an option offered with the lowest energy choice (e.g., artificially sweetened beverage (ASB), lowest calorie sandwich filling) and an option with the highest energy choice (e.g., sugar sweetened beverage (SSB), highest calorie sandwich filling) were created.

In total, 176 meal combos from nine fast-food chains were identified, which were allocated to one of 20 combo categories. The number of combos within a category ranged from three (5 categories) to 20 (4 categories).

##### Outcomes

A range of indicators were chosen to benchmark the healthiness of fast-food items and meal combos in relation to existing relevant targets and daily population recommendations for energy, sodium, added sugar and saturated fat intakes [24,25]

Serving size and nutritional composition/serving: The average amount of energy (kilojoules-kJ), sodium (mg), sugar (g) and saturated fat (g) per serving was calculated and described as medians (interquartile range); and minimum and maximum values. We calculated medians for these metrics because for some of the fast-food categories and combos, the nutrient contents were not normally distributed.

Percentage of adult daily recommendations/serving: The percentage contribution that each product and combo meal made to daily population Nutrient Reference Values (NRVs) [24] for energy, sodium, sugar and saturated fat was calculated. New Zealand and Australia share the same NRVs so we used the same energy benchmark as an Australian fast-food supply analysis of 8700 kJ [26]. We applied the recommended upper limit for sodium (2000 mg) [24] which is also the WHO upper limit [27], saturated and trans-fat (≤10% of energy/day, 23 g for 8700 kJ diet) [24] and sugar (WHO recommendation for free sugars intake ≤10% of energy/day, 51 g for 8700 kJ diet) [25]. The free sugars content of fast foods is not available and intrinsic sugars are present in few fast foods in NZ, therefore total sugars were used as a proxy.

Proportion of products exceeding the UK sodium targets: UK sodium targets (2024 targets published in 2020) were used as there are no targets available for NZ. UK sodium targets are set per serving (or slice for pizza) for the takeaway sector [23] and per 100 g for other categories (used for ‘cakes, muffins and pastry’), and were assessed for seven of 16 categories included in this study.

Proportion of sugar-sweetened beverages exceeding sugar thresholds for the UK soft drinks industry levy: The UK SDIL [16] is based on total sugar content of beverages per 100 mL. The UK soft drinks levy thresholds (>5 g and ≤8 g/100 mL and >8 g/100 mL) were used as NZ does not have sugar targets for beverages.

##### Analysis

All analyses were performed using SPSS software (version 25, IBM SPSS Statistics).

## 3. Results

### 3.1. Fast-Food Categories and Products

Table 1 presents the median (interquartile range) serving size, energy and nutrients/serving (sodium, total sugar, saturated fat), and percent contribution to daily recommendations by fast-food category. Appendix A presents, for each fast-food category, the minimum and maximum values for energy and nutrient contents and for percent contributions of energy and nutrients to daily recommendations.

The categories with the highest median energy per serving and percentage contribution to recommended daily energy intake (8700 kJ) were Burgers (2585 kJ, 29.7%, respectively), followed by Fries (2010 kJ, 23.1%) and Asian meals (2015 kJ, 23.2.0%). Half of the Pastry, savory, Cakes, muffins and pastries and Milkshakes/smoothies, considered to be snack items, contributed, respectively, at least, 22.0%, 22.4% and 18.6%/serving to the daily recommended energy intake for a NZ adult (Table 1).

The categories with the highest median sodium per serving and percentage contribution to maximum recommended daily sodium intake (2000 mg) intake were Burgers (1090.6 mg, 54.5%, respectively), followed by Breakfast, savory (1075 mg, 53.8%,) and Sandwiches and wraps (900 mg, 45.0%).) The categories with the highest median total sugar per serving and percentage contribution to maximum recommended daily free sugar intake (51 g) were Milkshakes/smoothies (49.0 g, 96.0%, respectively) followed by sugar-sweetened soft drinks, (33.8 g, 66.3%) and Cakes/muffins/pastries (32.8 g, 64.3%). The highest median values of saturated fat per serving and percentage contribution to maximum daily recommended intake (23 g) were observed for Pastry, savory (13.0 g, 56.8%, respectively) followed by Breakfast, savory (9.5 g, 41.3%) and Burgers (9.2 g, 40.0%) (Table 1).

Analysis involving all products showed that most (n = 1562; 89.1%) contributed 30% or less/serving to the daily recommended energy intake for an average NZ adult. In total, 235 products (13.4%) contributed 50% or more/serving of the maximum daily recommended sodium intake. Twenty-three products (1.3%) exceeded the maximum recommended daily sodium intake (data not shown in table).

Figure 2 shows the proportions of products above the UK sodium target [24] among the fast-food categories where a target existed. Overall, almost half the products (46.5%) exceeded the target/serving. The categories with the largest percentage of products exceeding the UK sodium target were Fries (100% of products), Pizzas (57.1%) and Pastries, savory (52.6%). The categories with the lowest percentage exceeding the UK sodium target were Sandwiches and wraps (38.1%), Burgers (36.5%) and Cakes, muffins and sweet pastry products (35.8%).

Of the 73 sugar-sweetened soft drinks assessed, the majority (n = 67; 91.8%) exceeded the UK-SDIL thresholds [16], where 58 (74.5%) had a sugar content >8 g/100 mL and 9 (12.3%) had a sugar content >5 g & ≤8 g/100 mL (data not shown). 

### 3.2. Fast-Food Combo Meals

Table 2 presents the medians (interquartile range) for energy and nutrients per combo meal (serving for one person) as well as the median percent contributions of energy and nutrients (sodium, total sugar, saturated fat) to daily recommendations. Appendix A presents, for each combo meal, the minimum and maximum values for energy and nutrient contents and for percent contributions of energy and nutrients to daily recommendations.

Overall, the median energy content of the combo meals/serving was 4381 kJ, meaning that 50% of the combo meals were contributing at least 50% of the average energy intake recommended for a NZ adult (8700 kJ). The combo meal categories with the highest median energy/serving and median energy contribution to daily recommended intake/serving were ‘Burger(s), fries, dessert, SSB’ (7531 kJ, 86.6%, respectively) and ‘Burger(s), fries, dessert, ASB’ (6463 kJ, 77.7%). The combo categories with the highest median energy/serving were those that contained more items and/or a dessert, and a sugary drink. The combo category with the lowest median energy/serving was ‘Burger or chicken and fries’ (2780 kJ, 32.0%). The combo categories with lower median energy were those constituted only by burger (or chicken) and fries and combos containing an ASB, sandwich or salad (Table 2).

Overall, the median sodium content of the meal combos was 1771.0 mg/serving, which corresponds to 88.6% of the maximum recommended daily sodium intake for adults (2000 mg) [24]. The meal combo categories with the highest median sodium/serving and highest median sodium contribution to daily maximum recommended intake/serving were ‘Chicken, potato, fries, additional item, ASB (2852.7 mg, 142.6%, respectively), ‘Chicken, potato, fries, additional item, SSB’ (2830.2 mg, 141.5), ‘Chicken, fries or potato, dessert, ASB’ (2353.5 mg, 117.47%) and ‘Chicken, fries or potato, dessert, SSB’ (2339.5 mg, 117.0%). The meal combo category with the lowest median sodium/serving was ‘Salad or wrap, smoothie’ (398.0 mg, 19.9% of daily maximum recommended intake). The meal combos with lower median sodium/serving were those containing regular rather than large serving sizes, combos with fewer items such as ‘burger, fries and drink’ and ‘chicken, fries and drink’ and combos based on sandwiches, pizza, pies, salad, or wraps (Table 2). Overall, among all meal combos examined, 84.1% (n = 148) contributed 50% or more of the maximum daily recommended sodium intake (data not shown).

Overall, the median total sugar content of the combo meals was 41.3 g/serving, corresponding to 81.0% of the maximum recommended intake of free sugar (51 g) [25]. Most (160/176) combos included a beverage option and those containing an SSB had the highest median total sugar per serving in relation to combos containing an ASB (Table 2). Overall, among all meal combos examined, one in three exceeded the WHO [25] maximum recommended intake of free sugars (data not shown).

Overall, the median saturated fat content of the combo meals was 10.6 g, corresponding to 46% of the maximum recommended intake (23 g) [24]. The combo meal categories with the highest median saturated fat/serving and median saturated contribution to daily maximum recommended intake/serving were ‘Burger(s), fries, dessert, SSB/ASB’ (both 19.1 g, 83.0%), ‘Chicken, fries or potato, dessert, SSB/ASB’ (both 17.3 g, 75.0%) and ‘Breakfast, hot chocolate (17.3 g, 75.2%). The fast-food combo meal category with the lowest median saturated fat/ serving was ‘Sandwich, chips, SSB/ASB’ (both 4.1 g, 17.6%) (Table 2). Overall, among all meal combos examined, one in ten exceeded the maximum daily recommended intake [24] for saturated fat.

## 4. Discussion

### 4.1. Summary of Findings

For NZ fast-food products for which nutrition information could be sourced (30%), many product categories and meal combos were high in energy and sodium and within some categories many products were high in total sugar and saturated fat. For products with a relevant UK sodium target, almost half exceeded the target. Over 90% of sugar sweetened soft drinks available in NZ fast-food outlets would be liable for the UK soft industry drinks levy. Burgers in particular had a high energy, sodium and saturated fat content.

The meal combos usually replace one of three usual main meal occasions in a day. However, half of the combos examined provided at least 50% of the daily energy requirement and 89%, 81% and 46% of the maximum recommended intake for sodium, sugar, and saturated fat. There was a wide range of combo options, and some provided a choice of product options, particularly sides or drinks. Combos with fewer items or smaller burgers, no dessert and an ASB (rather than a sugary drink) provided less energy, sodium, and sugar. For example, the median sugar and energy of the ‘chicken, fries and drink’ combo category was 49.2 g and 4090 kJ when it contained a sugary drink but only 1.5 g and 3286 kJ when it contained an ASB. The options based on sandwiches/wraps also tended to have less energy and sodium although there were not many of these combos.

### 4.2. Comparisons of Findings to Previous Studies

These findings are consistent with assessments of the fast-food supply in other countries including Australia, the US and Canada [26,28,29]. The Australian report on The State of the Fast-Food Supply in 2019 [26] concluded that most products were unhealthy, sold in oversized portions, and high in salt, sugar and harmful fats. The authors also commented on the lack of nutrition information with over half of Australian chains not providing sufficient data. In the United States, combination meals at chain restaurants were high in energy, sodium, saturated fat and sugar and most default options in meal combos exceeded national guidelines for calories and sodium [28]. In Canada, meals from fast-food chain restaurants were high in saturated fat, sodium and sugar [29]. An analysis of combo meals offered by quick-service restaurants in Australia [30] also found many combos provided more than 30% of an adult’s average daily energy intake. An earlier analysis conducted in NZ reported mean serving size, energy per serving and sodium per serving [13]. While we cannot directly compare with the 2016 NZ data, we found that Burgers and Asian meals were still in the top three categories for energy per serving, and burgers and sandwiches/wraps were still in the top three for sodium content per serving.

Of particular concern were the high sodium levels of many products, with many exceeding the UK benchmarks and almost half the combos exceeding the daily recommended maximum sodium intake. However, for every category where a benchmark existed, except for fries, there were also products that did not exceed that benchmark. This indicates that in most cases, it is possible to offer lower sodium options. There are no recent data to indicate the contribution of fast-food to New Zealanders’ sodium intake. However, a survey conducted in 2012 indicates that New Zealanders consume considerably more sodium (3373 mg) [31] than the Suggested Dietary Target (SDT) for NZ adults of 2000 mg/day [24]. One NZ study estimated that the mean daily sodium intake from savory fast foods for regular fast-food consumers was 1229 mg/day [32]. Another NZ study estimated the percentage contribution of sodium from takeaway and restaurant foods at 887 mg/day, 26.3% of sodium sources in NZ diet [33] As fast food consumption is growing, this contribution is likely to be higher now.

While the fast-food industry has grown, it appears that it has done little to improve the overall healthiness of the fast-food supply despite the recommendations made by the Food Industry Taskforce convened by the Ministers of Health and Primary Industries in 2018 [18]. Eyles et al. 2018 [13] found moderate to large increases in product serving size, and energy and sodium per serving from 2012 to 2016. An Australian analysis [26] that looked at changes in categories, rather than individual products found there was little change in the healthiness of products between 2016 and 2019.

### 4.3. Strengths and Limitations

A strength of this study is the systematic data collection from a large number of NZ fast-food chains, covering at least 60% of the fast-food sales [12], however data collection did not include small chains and independent retailers. Combination meals have not been assessed in NZ before and are useful to analyze as they provide the context of a meal when benchmarking against daily recommendations. As combination meals involve several options, the analysis of combo meals was carefully conducted; with two options analyzed for most combos: the healthiest (less energy and sugar) and the least healthy. There are also some important limitations to consider. Most NZ fast-food chains did not provide nutrition information on their products, so it was impossible to undertake a comprehensive analysis of the nutritional state of the national fast-food supply. Half of the chains that provided some nutrition information were international chains and half were national chains. However, of the six chains that did not provide any nutrition information about their products, five were international chains. There is also a chance that the products from chains that provided some nutrition information may have had a better nutrition profile in relation to products from chains that did not provide any information. This means that this study may have underestimated portion size and overestimated the healthiness of the fast-food supply. This study`s data were not sales weighted and therefore do not reflect the healthiness of items by frequency of consumption, though our analysis does include the most commonly consumed fast foods (bread-based dishes, fries, non-alcoholic beverages, poultry) in New Zealand [22].

### 4.4. Implications and Recommendations

This research highlights the need for policies, guidelines, and targets to improve the healthiness of fast food and provision of nutrition information as these do not currently exist in NZ. The Government needs to ensure the Food Industry Taskforce acts on the taskforce recommendations, and provide leadership by setting guidance for serving sizes, maximum targets for sodium content that are specific and measurable, and requiring fast-food outlets to provide nutrition information. A systematic review [14] found no standardized assessment methods or metrics to evaluate transnational chain restaurants’ practices to improve the healthiness of menu items. Public health experts recommend a robust, independent regulatory system with targets set by government and regular monitoring [34].

The wide range of serving sizes within food categories in fast-food outlets makes it difficult to compare products, apply benchmarks, and for consumers to choose healthier options and appropriate serving sizes. For example, in this study the serving size ranged from 43–298 g for pizzas and from 79–513 g for fries. Advice should include maximum energy values for combos, given that some were very high in energy. A scoping review found some expert-recommended targets for restaurants to improve products, but no internationally accepted standard for serving sizes [14]. In the US, some organisations provide targets for serving sizes for healthy meals such as the Healthy Restaurant Meal Standards [35] and the Heart-Check certification [36]. The Australian Healthy Food Partnership had a portion size working group (now disbanded) [37] to develop recommended portion sizes, including for fast food, and published targets for some nutrients and limited fast-food categories [38]. The UK has calorie reduction guidance for the eating out of home sector [39].

The substantial amount of sodium in the fast-food supply and the increasing consumption of takeaways in NZ warrants reformulation of fast foods to lower sodium and monitoring of the sodium content of fast-food products [40]. Excess sodium intake is a major preventable risk factor for hypertension [41], a leading cause of heart disease and stroke in NZ [1]. Few countries have targets for out-of-home foods. The UK sodium reduction targets are government-led, though voluntary but are regularly monitored [23]. WHO have recently published global sodium benchmarks but only a small number are applicable to fast food [42]. In addition, warning labels should be placed on those products and combos that exceed sodium targets. In 2015 New York City passed a sodium warning label rule, requiring chain restaurants to add a salt shaker icon beside menu items or combos containing more than 2300 mg of sodium [43].

Consumers have a right to know what is in their food and menu nutrition labeling is a strategy to provide this and can also encourage reformulation. Research in the U.S. suggests that the 2010 national menu labeling law may have influenced chain restaurants to reduce the energy content of newly introduced items [44,45,46]. Menu labeling is under consideration by Food Standards Australia NZ, with a consultation conducted in 2021 on a range of options for labeling the energy content of foods on the menu, including voluntary and mandatory options [47]. Menu labeling, particularly for energy content, is mandatory for fast-food chains in some countries such as Australia (5 jurisdictions) [47], Canada [48] and the U.S. [49] and will be mandatory for large businesses in the UK from 2022 [50].

Other areas that could improve the healthiness of fast-food menu offers include reformulation to reduce saturated fat and sugar content across menu items, healthier items (such as ASBs) to be the default option in combos and deals, introduction of healthier menu items, and marketing and pricing strategies to encourage purchasing of healthier items.

## 5. Conclusions

Nutrition information was available for one-third of products of major fast-food chains in New Zealand, limiting the generalizability of findings for the whole NZ fast food supply. Among products with information available, the majority had a high median content of energy and sodium. Some fast-food product categories had a high median content of sugar and saturated fat. Many serving sizes were large and varied considerably within a category. The majority of fast-food combo meals/serving provided a considerable contribution towards the daily recommended energy intake and the maximum daily sodium and sugar intake recommendations. This is the first comprehensive study of fast-food combo meals in NZ. This research benchmarks the current healthiness of the fast-food supply providing evidence to encourage Government to: (i) develop policy to ensure that NZ fast-food chains make nutrition information on their products readily available and, (ii) implement government-led guidance on serving sizes for fast foods including combos and (iii) set targets for sodium and sugar content, including warning labels for products that exceed such targets.

## Figures and Tables

**Figure 1 nutrients-13-04010-f001:**
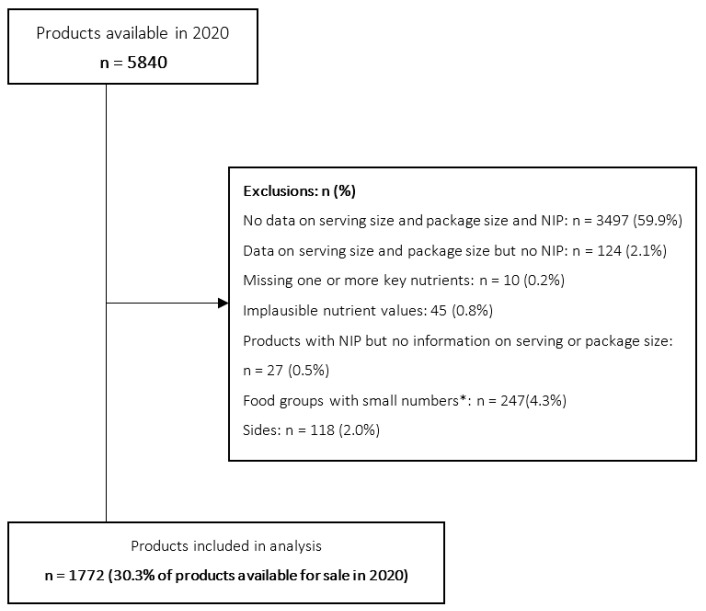
Flow-chart indicating the reasons and number of fast-food products excluded and included in the analyses, New Zealand, 2020. * Asian—Chinese, juice, beverages other, tea/coffee/hot chocolate, water, breakfast sweet, dressings/condiments sweet, other, seafood, soups.

**Figure 2 nutrients-13-04010-f002:**
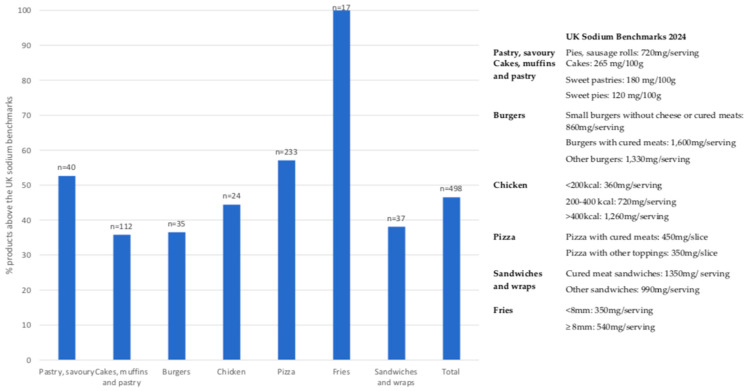
Proportions of products above the 2024 UK sodium targets for the respective food category.

**Table 1 nutrients-13-04010-t001:** NZ fast food supply 2020, by food category: Median (interquartile ranges) for: total energy, sodium, sugar and saturated fat content by serving size and percent energy, sodium, sugar and saturated fat contributions to recommended daily intakes.

Major and Minor Fast-Food Category	Serve Size (g/mL)		Energy and Nutrients								
				Energy—Kilojoules/Serving		Sodium—Milligrams/Serving		Total Sugar—Grams/Serving		Saturated Fat—Grams/Serving	
	N	Median (IQR)	N	Content	% of Daily Recommendation *	Content	% of Daily Recommendation *	Content	% of Daily Recommendation *	Content	% of Daily Recommendation *
				Median (IQR)	Median (IQR)	Median (IQR)	Median (IQR)	Median (IQR)	Median (IQR)	Median (IQR)	Median (IQR)
Asian (sushi, katsu chicken, noodles)	47	302.0 (267.0–322.0)	48	2015.0 (1794.6–2280.0)	23.2 (20.6–26.2)	673.0 (535.5–1028.8)	33.7 (26.8–51.4)	12.2 (11.2–13.2)	23.9 (22.0–25.9)	2.1 (1.2–3.6)	9.0 (5.0–15.8)
Beverages											
Milkshakes, smoothies	99	475.0 (402.0–480.0	130	1617.0 (1220.0–2519.5)	18.6 (14.0–29.0)	154.5 (100.0–252.6)	7.7 (5.0–12.6)	49.0 (39.5–70.6)	96.0 (77.5–138.5)	6.3 (2.7–10.7)	27.4 (11.6–48.6)
Soft Drinks, sugar sweetened	51	328.0 (250.0–375.0)	72	590.5 (457.0–774.0)	6.8 (5.3–8.9)	27.3 (12.1–55.3)	1.4 (0.6–2.8)	33.8 (27.0–45.2)	66.3 (52.9–88.5)	0.0 (0.0–0.0)	0.0 (0.0–0.0)
Soft Drinks, artificially sweetened	15	330.0 (330.0–355.0)	24	8.5 (5.0–35.8)	0.1 (0.6–0.4)	32.5 (5.0–53.8)	1.6 (0.3–2.7)	0.0 (0.0–0.0)	0.0 (0.0–0.0)	0.0 (0.0–0.1)	0.0 (0.0–0.3)
Breakfast, savory (egg dishes, bread dishes)	28	212.0 (167.3–245.8)	36	1920.0 (1552.5–2224.8)	22.1 (17.8–25.6)	1075.0 (758.0–1287.5)	53.8 (37.9–64.4)	3.2 (2.3–6.3)	6.3 (4.6–12.3)	9.5 (5.9–14.8)	41.3 (25.7–64.1)
Pastry, savory	87	200.0 (180.0–220.0)	88	1935.0 (1630–2225.0)	22.2 (18.7–25.6)	725.5 (604.3–910.8)	36.3 (30.2–45.5)	2.6 (1.3–4.0)	5.0 (2.6–7.8)	13.0 (9.3–16.2)	56.8 (40.2–70.3)
Cakes, muffins and pastry	296	150.0 (100.0–167.0)	315	1950.0 (1510.0–2454.0)	22.4 (17.4–28.2)	299.0 (223.0–366.0)	15.0 (11.2–18.3)	32.8 (24.7–42.2)	64.3 (48.4–82.7)	6.3 (4.2–10.1)	27.4 (18.3–43.9)
Desserts	60	105.5 (90.0–160.0)	75	1250.0 (833.0–1700.0)	14.4 (9.6–19.5)	133.0 (70.0–221.0)	6.7 (3.5–11.1)	26.5 (18.0–38.6)	52.0 (35.3–75.7)	7.5 (4.2–11.7)	32.6 (18.3–50.9)
Burgers	118	300.4 (197.5–373.0)	149	2585.0 (1999.5–3370.0)	29.7 (23.0–38.7)	1090.6 (746.5–1462.0)	54.5 (37.3–73.1)	6.7 (8.5–11.9)	16.7 (13.1–23.3)	9.2 (5.5–17.4)	40.0 (23.7–75.7)
Chicken	57	95.0 (52.0–200.0)	63	927.0 (421.0–1770.0)	10.7 (4.8–20.3)	594.0 (335.0–964.0)	29.7 (16.8–48.2)	0.5 (0.2–1.6)	1.0 (0.3–3.1)	2.0 (1.4–4.0)	8.7 (6.1–17.4)
Pizza	414	93.5 (74.0–126.0)	416	850.5 (694.5–1199.8)	9.8 (8.0–13.8)	433.5 (336.0–599.8)	21.7 (16.8–30.0)	2.6 (1.7–4.0)	5.1 (3.3–7.8)	3.3 (2.4–5.0)	14.5 (10.4–21.7)
Salads	54	266.3 (148.5–332.3)	59	916.0 (524.0–1230.0)	10.5 (6.0–14.1)	542.0 (350.0–758.0)	27.1 (17.5–37.9)	5.5 (3.1–7.4)	10.8 (6.0–14.5)	2.3 (0.9–4.2)	10.0 (3.9–18.3)
Sandwiches and wraps	101	192.5 (238.0–258.5)	113	1748.0 (1385.0–2165.0)	20.1 (15.9–24.9)	900.0 (660.5–1320.0)	45.0 (33.0–66.0)	5.4 (4.0–7.8)	10.6 (7.8–15.3)	5.8 (3.0–8.5)	25.2 (12.8–37.0)
Fries	22	246.0 (156.0–302.0)	25	2010.0 (1370.0–2790.0)	23.1 (15.7–32.1)	650.0 (284.0–1334.5)	32.5 (14.2–66.7)	0.7 (0.5–1.8)	1.4 (1.0–3.5)	2.6 (1.7–5.9)	11.3 (7.2–25.4)
Sides, other	45	120.0 (35.0–253.0)	52	867.5 (305.5–1677.5)	10.0 (3.5–19.3)	321.5 (69.3–682.5)	16.1 (3.5–34.1)	3.3 (0.5–8.0)	6.4 (1.0–15.6)	1.9 (0.7–5.0)	8.3 (3.2–21.7)
Dressings/condiments, savory	49	21.0 (16.0–30.0)	88	207.0 (135.3–371.0)	2.4 (1.6–4.3)	135.0 (88.8–194.5)	6.8 (4.4–9.7)	2.0 (1.0–5.9)	3.9 (2.0–11.6)	0.4 (0.0–1.0)	1.5 (0.0–4.3)

IQR: Interquartile ranges. * Percentage calculated having as reference the recommended adult average daily energy intake (8700 kilojoules/day), sodium intake (2000 mg/day), free sugars intake (maximum 51 g/day based on 8700 kJ), saturated fat intake (maximum of 23 g/day based on 8700 kJ). Missing (n) Serving size-g/mL: Asian (12); Milkshakes, smoothies (31); Soft drinks-sugar sweetened (22); Soft drinks-artificially sweetened (9); Breakfast-savory (8); Pastry-savory (1); Cakes, muffins and pastry (19); Desserts (15); Burgers (31); Chicken (12); Pizza (3); Salads (5); Sandwiches and wraps (12); Fries (3); Sides, other (7); Dressings/condiments-savory (39). Missing (n) Nutrient content and % of daily recommendation: Asian (11); Milkshakes, smoothies (0); Soft drinks-sugar sweetened (1); Soft drinks-artificially sweetened (0); Breakfast-savory (0); Pastry-savory (0); Cakes, muffins and pastry (0); Desserts (0); Burgers (0); Chicken (6); Pizza (1); Salads (0); Sandwiches and wraps (0); Fries (0); Sides, other (0); Dressings/condiments-savory (0).

**Table 2 nutrients-13-04010-t002:** NZ fast food supply 2020, by combo types: Median (interquartile ranges) for: energy, sodium sugar and saturated fat content by serving size and for percent energy, sodium, sugar and saturated fat contributions to recommended daily intakes.

Fast-Food Meal Combo Category	N	Energy— Kilojoules/Serving		Sodium—Milligrams/Serving		Total Sugar—Grams/Serving		Saturated Fat—Grams/Serving	
		Content	% of Daily Recommendation *	Content	% of Daily Recommendation *	Content	% of Daily Recommendation *	Content	% of Daily Recommendation *
		Median (IQR **)	Median (IQR **)	Median (IQR **)	Median (IQR **)	Median (IQR **)	Median (IQR **)	Median (IQR **)	Median (IQR **)
Burger(s), fries, drink-SSB	20	4042.0 (3574.5–4942.5)	46.5 (41.1–56.8)	1299.1 (1157.8–2000.5)	65.0 (57.9–100.0)	55.1 (40.2–62.7)	108.0 (78.8–123.0)	10.1 (7.1–14.8)	43.7 (31.0–64.2)
Burger(s), fries, drink-ASB	20	3380.0 (2805.1–4108.3)	38.9 (32.2–47.2)	1319.5 (1173.9–2023.0)	66.0 (58.7–101.2)	8.1 (6.9–11.5)	15.8 (13.6–22.5)	10.1 (7.1–14.8)	43.7 (31.0–64.2)
Burger(s), fries, dessert, drink-SSB	9	7531 (5985.8–8301.0)	86.6 (68.8–95.4)	2413.0 (1843.6–2550.0)	120.7 (92.2–127.5)	80.3 (71.4–89.5)	157.5 (139.9–175.5)	19.1 (17.9–30.2)	83.0 (77.6–131.3)
Burger(s), fries, dessert, drink-ASB	9	6763.0 (5433.3–7533.0)	77.7 (62.5–86.6)	2433.0 (1866.6–2570.0)	121.7 (93.3–128.5)	39.4 (33.4–43.9)	77.3 (65.5–86.1)	19.1 (17.9–30.2)	83.0 (77.6–131.3)
Chicken, fries, drink-SSB	3	4090.0	47.0	1492.0	74.6	49.2	94.5	8.3	47.0
Chicken, fries, drink-ASB	3	3286.8	37.8	1514.5	75.7	1.5	2.9	8.3	36.2
Chicken, fries or potato, dessert, drink-SSB	10	5957.0 (4588.1–7615.5)	68.5 (52.7–87.5)	2339.5 (1657.6–2948.0)	117.0 (82.8–147.0)	50.6 (42.7–63.5)	99.1 (83.8–124.4)	17.3 (10.0–21.3)	75.0 (43.4–92.5)
Chicken, fries or potato, dessert, drink-ASB	10	5404.0 (3839.4–6847.5)	62.1 (44.1–78.7)	2353.5 (1668.9–2968.0)	117.7 (83.4–148.4)	13.6 (7.3–17.9)	26.6 (14.2–35.0)	17.3 (10.0–21.3)	75.0 (43.4–92.5)
Sandwich, chips, drink-SSB	6	3789.9 (3438.8–4073.2)	43.6 (39.5–46.8)	1218.0 (800.3–1523.4)	60.9 (40.0–76.2)	53.3 (51.8–54.1)	104.4 (101.5–106.1)	4.1 (3.3–5.9)	17.6 (14.1–25.5)
Sandwich, chips, drink-ASB	6	2986.7 (2635.6–3270.0)	34.3 (30.3–37.6)	1240.5 (822.8–1545.9)	62.0 (41.1–77.3)	5.6 (4.1–6.4)	10.9 (8.0–12.6)	4.1 (3.3–5.9)	17.6 (14.1–25.5)
Pizza(s), Side(s)	13	3302.0 (2938.8–3892.0)	38.0 (33.8–44.7)	1033.0 (793.0–1270.0)	51.7 (39.7–63.5)	42.6 (10.1–46.9)	83.5 (19.7–91.9)	11.7 (9.9–13.4)	50.9 (43.0–58.3)
Pizza(s), Side(s), Drink-SSB	5	4517.0 (4352.0–4748.5)	51.9 (50.0–54.6)	891.0 (767.0–1200.0)	44.6 (38.4–60.0)	77.9 (74.8–82.6)	152.7 (146.6–161.9)	13.6 (11.1–15.9)	59.1 (48.3–69.1)
Pizza(s), Side(s), Drink-ASB	5	3927.6 (3762.6–4159.1)	45.1 (43.2–47.8)	894.3 (770.3–1203.3)	44.7 (38.5–60.2)	42.9 (39.8–47.6)	84.1 (77.9–93.2)	13.6 (11.1–15.9)	59.1 (48.3–69.1)
Pie, side (optional), drink-SSB	4	4062.0 (3727–4547.0)	46.7 (42.8–52.3)	1239.0 (952.5–1377.8)	62.0 (47.6–68.9)	49.6 (47.4–63.3)	97.3 (93.0–124.0)	13.9 (9.4–27.0)	60.4 (40.8–117.3)
Pie, side (optional), drink-ASB	4	3311.0 (2976.0–3796.0)	38.1 (34.2–43.6)	1260.0 (973.5–1398.8)	63.0 (48.7–69.9)	4.6 (2.4–18.3)	9.0 (4.8–35.8)	13.9 (9.4–27.0)	60.4 (40.8–117.3)
Breakfast, hot chocolate	3	4581.0	52.7	2163.0	108.2	29.1	57.1	17.3	75.2
Salad or wrap, smoothie	3	3051.0	35.1	398.0	19.9	50.6	99.2	6.1	26.5
Burger or chicken and fries	3	2780.0	32.0	1261.0	63.1	5.9	11.6	8.0	34.8
Chicken, potato, chips, additional item ***, drink-SSB	20	5816.1 (5433.5–6658.8)	66.9 (62.5–76.5)	2830.2 (2452.0–3173.7)	141.5 (122.6–158.7)	59.5 (58.0–65.7)	116.7 (113.8–128.8)	10.1 (7.7–12.6)	43.7 (33.5–54.8)
Chicken, potato, chips, additional item ***, drink-ASB	20	5012.0 (4630.3–5855.6)	57.6 (53.2–67.3)	2852.7 (2474.5–3196.2)	142.6 (123.7–159.8)	11.8 (10.3–18.0)	23.1 (20.2–35.2)	10.1 (7.7–12.6)	43.7 (33.5–54.8)
Total	176	4381.0 (3505.8–5660.9)	50.3 (40.3–65.1)	1771.0 (1172.5–2571.0)	88.6 (58.6–128.6)	41.3 (10.6–57.7)	81.0 (20.7–113.1)	10.6 (8.0–16.8)	46.1 (34.8–73.0)

IQR: Interquartile ranges. * Percentage calculated having as reference the recommended adult average daily energy intake (8700 kilojoules/day), sodium intake (2000 mg/day), free sugars intake (maximum 51 g/day based on 8700 kJ), saturated fat intake (maximum of 23 g/day based on 8700 kJ). ** Only median presented if n = 3. *** Burger, sandwich, bread roll or coleslaw. SSB: Sugar-sweetened beverage, ASB: Artificially-sweetened beverage.

## Data Availability

Because of the commercial and legal restrictions to the use of copy-righted material, it is not possible to share data openly, but unredacted versions of the dataset are available with a licensed agreement that they will be restricted to non-commercial use. For access to Nutritrack, please contact the National Institute for Health Innovation at the University of Auckland at enquiries@nihi.auckland.ac.nz.

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
