# Peer review of "Energy, Sodium, Sugar and Saturated Fat Content of New Zealand Fast-Food Products and Meal Combos in 2020"

_nutrients, 2021, doi:10.3390/nu13114010_

Round 1

Reviewer 1 Report

This study provides valuable evidence about the nutrient and energy content of fast-food meals available in NZ. The analysis focuses both on the nutrient and energy content of single items and of meal combos. The authors compared the nutrient composition of the surveyed products with international benchmarks. This makes the report very dense of information and difficult to understand in some parts. I would suggest some restructuring, better use of the subheadings and more table and graphs where appropriate so that the reader can easily and quickly understand the findings of the study. Moreover, I wonder what is the point of comparing drinks with the targets set by the SDIL if there is nothing restaurants could do to reduce the sugar content in their products.  I suggest deleting this analysis to better focus on the sodium and calorie content of these products as this is a more considerable and urgent problem. Please see below some other specific suggestions to improve the clarity of the analysis.

L2 remove full stop from the title

L33-L94-The introduction is quite lengthy and some sections should be deleted. I suggest to focus only on the nutritional quality of fast-food meals and products.

L37- The introduction could start from this line. The previous section could be deleted

L71- The UK gvt has also set calorie targets for fast-food items. Why did you not consider them? The link included in reference 18 does not seem to work

L72- This sentence on the SDIL pops out of the blue. In comparison to the mandatory nutrient and energy targets, the SDIL is a mandatory policy which prompts the drink industry to reformulate their products and not the fast-food sector to reformulate their drinks. I wonder what is the point of including sugary drinks in this analysis as there is nothing that restaurants could do to reduce the amount of sugar in the drinks they offer.

L79- I would specify “for fast food items”

L96- how many restaurants did you survey? What are the inclusion/exclusion criteria? In the abstract you wrote that you surveyed 27 restaurants. I could not find this information in the main text

L98 I suggest the following rewording “fast food chains were defined as restaurants providing food which is generally cheap, requires minimal preparation and where no table service is required”

L116 I suggest the following “ those sides  not available for sale separately were excluded from the analysis”

L122 Why considering only sugar content in drinks and not in solid food categories.

L109-112 The two sentences could be condensed into one sentence

L96- Could you use the active form here? For example “ we carried out a cross sectional survey”

L137- I would consider only the default choice (e.g. regular coke rather than artificially sweetened) as this is the most consumed option. What is the point of considering meals with the ASB in your analysis if everybody knows that these do not contain sugar and thus reduce the calories

L197 Please use a subheading here.

L197 – are there any other analyses looking at the sugar content of soft drinks in NZ? I feel that this category should be excluded from the present analysis and should be assessed separately (i.e. in another paper). Also, your analysis includes milkshakes. Why did you not use the UK PHE sugar targets for milkbased products? https://assets.publishing.service.gov.uk/government/uploads/system/uploads/attachment_data/file/984282/Sugar_reduction_progress_report_2015_to_2019-1.pdf

L193 Please consider deleting this sentence “there are few other internationally published fast- food targets, they are the most well-known, and they have been in existence for some time(first published in 2017)” as it is not very clear

L197-200- the UK has set sugar targets also for solid food categories (e.g. cakes). Why did you consider only the sugar content in drinks and not for solid food items?

L 219 what does the “Breakfast, savoury” include? I suggest rewording and be more specific. The same with Asian meals. What do consider an Asian meal in NZ? A curry and a side of rice? A bowl of noodles with vegetables?

Figure 2. The quality of the image is very low. Could you please fix the resolution (and possibly also the aesthetics) of this figure? For example by deleting the small blue square in the middle of the image as this provide redundant information. The caption for Figure 2 should mention the salt targets (not the generic term “UK targets”). By the way, are you using the most recent salt targets? The table on the right should have a heading specifying what are the values below. Why some products have salt targets per 100g and other as per serving?

L279-280: these sentences are not clear.

L309 this sentence is very clear

L315 the implication of this finding could be that restaurants should start offering ASB as default option as this lower the energy and sugar content of a meal

L317 I would also add that wraps and salads are very different food categories and that there is a wide variation of items within each category

L354 The implications section should be shorted and more straight to the point. I suggest shortening and deleting some parts

L350 could you explain here and earlier in the text what is the food industry taskforce?

L360- You could mention that in the UK calorie labelling will be mandatory from 2022 for larger businesses https://lordslibrary.parliament.uk/eating-out-and-takeaways-calorie-labelling-regulations/

L376 cut-offs for calories and portion size exists also in the UK ( see Eating out, takeaway and delivery sector in this document https://assets.publishing.service.gov.uk/government/uploads/system/uploads/attachment_data/file/915367/Calorie_reduction_guidelines-Technical_report_070920-FINAL.pdf)

L384-386 please consider deleting this sentence

L398  as previously mentioned calorie labelling will be mandatory in the UK from 2022

L402-408 please consider deleting this sentence

L410 are the terms fast-food restaurants and quick service restaurants the same?

I would also add in the discussion that the fast-food chains and restaurants surveyed might have a better nutrient profile just because they make the nutrient information somewhat available to customers. Restaurants not making this information available might have larger portion size and higher levels of unhealthy nutrients in their meals.

Reviewer 2 Report

Dear Authors,

The manuscript (nutrients-1433297), presented for review, is very interesting.

Authors, Please note and address the following comments:

Abstract

The authors relate the percentages to different product groups, which in my opinion, makes the work unclear. Examples are given below.

Only 1,777 fast-foods were analyzed, and in the Abstract section, this number refers to the entire group of fast food available in New Zealand in 2020, i.e., n=5,840 (Figure 1), which stated 30.4% (lines 21-22). On the other hand, the percentage of 91.8% appears in the next sentence with the number (n=67) sugar-sweetened drinks. It is completely incomprehensible to me.

Material and methods

Lines 96-99: In my opinion, this information should be rather presented in the Introduction.

It would be interesting if the authors revealed how many of the analyzed fast food establishments are international chains. This is important as e.g. McDonald's, also present in New Zealand, has nutrition information on its website. I checked that this is information about energy, protein, fat, carbohydrates, sugars, and sodium, similar to the other countries in the world. In addition, the company has a policy of change aimed at reducing simple sugars and sodium in meals, currently at Happy Meals. Information on how many of these assessed products came from fast-food chains, including international chains, New Zealand-specific fast-food chains, and how many local fast food were there. It is interesting and could be part of the discussion.

The authors in the methodology explained how many fast food products were analyzed. However, in the supplementary material, if we sum up the number of products assessed, it is 1753, and in Table 2 - 114 combo meals. This gives 1,867 assessed products. The authors write that 116 products have been repeated.

I tried to check this number, and I  don't know why it is 1777 and not 1753 or 1867. For example, in Figure 2, which shows the products rated for salt, the total number of products is 498. Figure 2 is understandable.

On the other hand, when I add up the number of beverages analyzed in Table 1 (lines 235-236), there are 165 of them, and in Table 2 - 176 other products. So it is not clear to me where the number 1777 came from.

4.1. Implications and recommendations – This section is very well written.

Conclusions

Lines 425 – This sentence is not fully related to the results: „The majority of foods in the NZ fast food supply were high in energy and sodium and within some categories, products were high in sugar and saturated fat”.

The authors removed many products from the analysis due to the lack of necessary information about the products, due to this they can't write "the majority of products in New Zealand". If the authors write such a sentence, in my opinion, information about the number of analyzed products should be added.

Despite my comments, I am pleased to recommend this manuscript for publication. I believe that it concerns an important area of research in an international context. This information is important for New Zealanders and their health, as well as for other researchers to compare their results across countries.

Reviewer

Round 2

Reviewer 1 Report

Dear editorial office,

I read the revised version of the manuscript and I am satisfied with the changes made by the authors. I am also happy about all the clarifications provided in the rebuttal letter. The manuscript has been substantially improved and in my opinion, this is now ready to be published in Nutrients.

Author Response

Thank you very much for taking time to review the revisions.